

WIND
ENERGY
SCIENCE

# System identification, fuzzy control and simulation results for a fixed length tether ~~of a~~ kite power system

**Tarek N. Dief[1], Uwe Fechner[2], Roland Schmehl[3], Shigeo Yoshida[4], Amr M. M. Ismaiel[1,5], and Amr M. Halawa[1]**

[1]Department of Earth System Science and Technology, Kyushu University, Fukuoka, Japan
[2]Aenarete – Wind Drones, The Hague, the Netherlands
[3]Delft University of Technology, 2629 HS Delft, the Netherlands
[4]Research Institute for Applied Mechanics, Kyushu University, Fukuoka, Japan, Japan
[5]Faculty of Engineering and Technology, Future University in Egypt (FUE) TS1, Egypt

**Correspondence:** Tarek N. Dief (tarek.na3em@riam.kyushu-u.ac.jp)

**Abstract.** TS2 In wind energy research, airborne wind energy systems are one of the promising energy sources in the near future. They can extract more energy from high altitude wind currents compared to conventional wind turbines. This can be achieved with the aid of aerodynamic lift generated by a wing tethered to the ground. Significant savings in investment costs and overall system mass would be obtained since no tower is required. To solve the problems of wind speed uncertainty and kite deflections throughout the flight, system identification is required. Consequently, the kite governing equations can be accurately described. In this work, a simple model was presented for a tether with a fixed length and compared to another model for parameter estimation. In addition, for the purpose of stabilizing the system, fuzzy control was also applied. The design of the controller was based on the concept of Mamdani. Due to its robustness, fuzzy control can cover a wider range of different wind conditions compared to the classical controller. Finally, system identification was compared to the simple model at various wind speeds, which helps to tune the fuzzy control parameters. TS3

## 1 Introduction

Airborne wind energy (AWE) systems are very promising energy sources that use flying devices. These devices can fly at high altitudes. Therefore, power can be generated by harvesting stronger and more persistent wind. The kite system is one of the AWE systems being developed. It consists mainly of two parts, a flexible wing and a generator on the ground connected by a tether. To capture as much power as possible from wind, the kite should fly at a high crosswind speed. To satisfy this, control is applied to the kite to keep it flying at high altitude, and perpendicular to the direction of the wind in an optimized path (Fagiano and Milanese, 2012; van der Vlugt et al., 2013).

AWE systems can capture more energy with higher capacities, which is why they are considered a good renewable energy system. The wind energy density at an altitude of 10 km could reach up to 5000 W m$^{-2}$ according to Wubbo Ockels, the developer of the "ladder-mill" concept in 1997 (Ockels, 2001). However, it is ~~too hard~~ <sup>technically too challenging</sup> to build a system that can operate at an altitude of 10 km and generate electricity from wind. This is why most of the current development and research projects have shifted their focus to lower altitudes (Archer et al., 2014).

Wind energy density ranges from 1400 to 4500 W m$^{-2}$ at altitudes of 200–900 m. Wind turbines cannot be installed at these altitudes because of the limitations of tower size (Goudarzi et al., 2014). Therefore, it would be an optimum solution to have a system similar to wind turbines at this altitude but with no tower. The concept of wind turbine blade rotary motion can be replaced by a ~~tethered~~ kite connected via the flexible ~~wing~~ <sup>tether</sup> to a fixed generator on the ground.

The power generation by AWE follows several concepts; however, in this paper we have only mentioned two different concepts. The first concept is based on the tension force in the tether. The flying wing pulls the tether, which is wrapped around a ~~pulley~~ on the ground connecting it to the generator, until the tether reaches its maximum length. Then, it is reeled back to the minimum length allowed based on the design limitations. The second concept depends on installing ~~a motor and generator setup~~ on the wing itself, which generates energy during ~~most of the cycle and uses energy during the other part of it.~~ It sends the generated energy through the electrified tether to the ground. It is crucial for the kite system to control its motion for efficient and reliable operation.

The optimum trajectory for kite flight is one of the key control parameters that can be decided by a flight path planner. To keep the kite on this planned trajectory, a winch controller controls the tether length. The kite flight has two main phases, as shown in Fig. 1. First, the reel-out phase is where the kite is free to go further from the ground station and pulls the tether. To obtain the maximum tensile force, the angle of attack of the wing is maximized. Second, the reel-in phase is where the kite is pulled back toward the ground station. In this case, the angle of attack is minimized to reduce the drag force on the kite, which would cost more energy.

Many researchers have studied the control of the kite system (Canale et al., 2010; Jehle and Schmehl, 2014b; Ilzhöfer et al., 2007; Baayen and Ockels, 2012; Williams et al., 2008; Houska and Diehl, 2007; Costello et al., 2013; Diehl et al., 2001; Fagiano et al., 2014; Erhard and Strauch, 2013). However, they only considered the first phase of the kite motion, which is considered for power generation, and neglected the second phase, where energy is used to pull the kite back. Other studies were concerned with the modeling of the kite system, winch controller and tether assembly (Diehl, 2001; Ahmed, 2014; Fagiano, 2009; Furey, 2012; Thorpe, 2011; Zgraggen, 2014). The governing equations in most of these studies were defined by using the point mass model (Fechner et al., 2015). Other researchers considered the governing equations based on a rigid body model without considering the turn rate law, which is necessary to describe steering of the kite (Thorpe, 2011; Zgraggen, 2014; Fechner et al., 2015; Williams et al., 2007). Other researches discretized the kite into 10 points, which increased the solution accuracy, although the tether was not discretized (Furey, 2012).

Neural network modeling was an idea that was analyzed, but the results were not satisfactory. Quasi-static modeling was also considered for more accurate controller implementation, but the results were not sufficient for validation (Fagiano et al., 2012; Erhard and Strauch, 2013). The average system model overcomes this validation problem since it gives a suitable derivation for different types of controllers (Fechner and Schmehl, 2012).

Experimental efforts for the autonomous take-off of the airborne systems were carried out, however there are still some challenges to get fully autonomous flight in differ-

ent wind conditions. Moreover, a global controller that can work under all conditions cannot be designed effectively for the commercial products (Fechner and Schmehl, 2012; Jehle and Schmehl, 2014a; Baayen and Ockels, 2012). Nonlinear model predictive control (NMPC) is used as a control strategy by many researchers to stabilize the kite. It is possible to theoretically apply this algorithm to optimize flight trajectory, but in a real flight test, it will require accurate and fast wind data that are currently unavailable (Canale et al., 2010; ~~Jehle and Schmehl, 2014b;~~ Ilzhöfer et al., 2007).

Thus, alternative techniques are needed to stabilize the flight trajectory. One technique is very promising for fixed short tethers, as it does not require information about the wind field or the kite and still performs quite well (Fagiano et al., 2014). Neither long nor variable length tethers are valid for the simulation. For a tether with a length of 200–500 m, or for a heavy kite, the accuracy is insufficient. Accuracy was increased in other studies that considered the apparent wind speed and gravitational effects in the simulation (Jehle and Schmehl, 2014a). However, for a tether shorter than 200 m with a time delay greater than 200 ms, the accuracy becomes insufficient.

The uncertainty of the kite's models has recently been presented in Fagiano et al. (2014); Jehle and Schmehl (2014a); Fagiano and Milanese (2012); van der Vlugt et al. (2013). However, several practical questions arise when dealing with the control design process. It is crucial to identify the wind speed, direction, aerodynamic parameters, kite shape and tether shape in real time. Thus, fully autonomous flight for the kite system has not yet been successful.

In this paper, the least square estimation (LSE) was used as a system identification to get a more accurate description for the steering dynamics of the kite in real time; the characteristics of the kite are varying with time because the wing is inflatable and flexible. Also, the wind speed can not be measured in real time, thus it is impossible to obtain the lift and drag forces during flight. This technique especially is used to identify the system parameters as it can calculate them analytically ~~with~~out iteration ~~(one directional calculations)~~ which means ~~no time loses~~ short calculation times and low chance of singularity in the solver.

The novelty of this work is to use an algorithm that is valid for any kite size and any tether length, so it can overcome the problems of the uncertainty. The LSE algorithm needs the steering values from the motors and the course angle from the sensors. Thus, no additional information is needed, such as the wind speed or the mathematical model of the kite, to identify the system that shall be controlled. Therefore, this paper tries to stabilize the kite using fuzzy control based on the LSE in real time.

This paper is divided into five main sections. The first section is the introduction, which gives an overall view of the previous research related to the paper's work. The second section shows the mathematical model (Sect. 2) used to describe the kite's motion. The third section gives the system identification derivation and details the sequence of the code

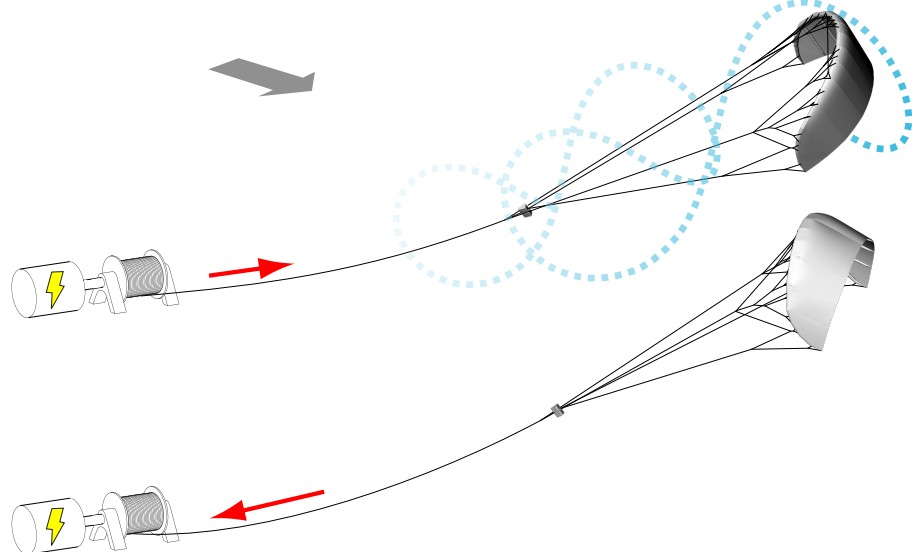

**Figure 1.** Working principle of the pumping kite power system (van der Vlugt et al., 2013). TS4

(Sect. 3). The fourth section describes the main parts of the fuzzy control and explains the choice of the fuzzy control parameters (Sect. 4). Finally, the last section shows the simulation results of the classical control and the fuzzy algorithm. The comparison also includes varying wind conditions and their effects on system stability (Sect. 5).

## 2  Mathematical model

Different mathematical models have been used to derive the kite governing equations. Some of these models considered the system as a kite connected with two control tethers without considering the variation of the angle of attack. These assumptions were considered to allow for an easier implementation of the kite's dynamic states (Diehl et al., 2001). Then, more complexity and details were added to the models. Some researchers have assumed that the system consists of the three degrees of freedom model and that the kite is a point mass at the end of a straight tether. Furthermore, the aerodynamic properties of the kite were considered fixed for all wind conditions (Ahmed et al., 2011).

Recent work considered the kite with a variable tether length and started to derive mathematical models for this variation of the tether length. A discretized tether model was derived during the reel-in and the reel-out phases using the Lagrangian approach to obtain the governing equations. Moreover, this model considered the segments of the tether as a rigid body connected by spherical joints (Williams et al., 2007).

Other research groups considered the tether model as a discretized tether with point masses connected by springs to each other, and aerodynamic analysis was performed using the vortex lattice method; however, the phases of reeling-in and reeling-out were not mentioned in the analysis (Gohl and Luchsinger, 2013). On the other hand, other research groups detailed the reel-in and reel-out phases (including the winch model) to present full kite motions in different phases (Ahmed et al., 2011; Coleman et al., 2013).

Some studies on kite design are being conducted to assess the aerodynamic characteristics. They applied the fluid-structure interaction method to study the aero-elasticity of the kite since the kite consists of an inflatable wing (Viré, 2012; Viré et al., 2012; Bosch et al., 2014). However, the simulation is slower than the real flight test, and it still needs more work to run the same as the real flight test.

Recent work on kite modeling has been achieved at TU Delft by Fechner et al. (2015). This research considered the dynamics of all system components such as the tether, kite, and generator. Additionally, the reel-in and reel-out phases were detailed to provide a smooth simulation for the tether. Additionally, the authors used two different definitions for the kite. The first definition considered the kite as an improved point mass model. It can be used to calculate the angle of attack and calculate the lift and drag of the kite during changes in the angle of attack. The other model considered the kite as a four-point mass model with rotational inertia in all axes. It closely models the real kite since all dimensions of the kite (the height and the width) are considered.

This section is divided into three main subsections. The first subsection presents the system model and gives a full description for the kite kinematics framework (Sect. 2.1). The second subsection explains the flight path planner (FPP) (Sect. 2.2) to show the kite path during flight and to show the parameters that affect the kite trajectory. The FPP was chosen to adapt any testing area for the flight test. Finally, the flight path controller (FPC) (Sect. 2.3) was derived to sta-

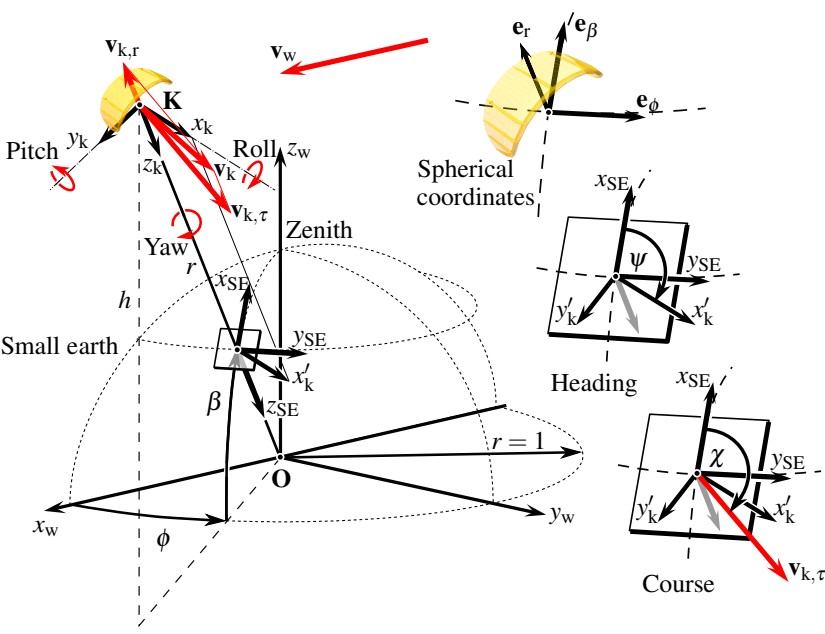

**Figure 2.** *Small Earth reference frame*: the position of the kite is projected onto a half sphere with a radius of one. The elevation angle $\beta$ and the azimuth angle $\phi$ describe the position of the kite, the angle $\psi$ ~~and~~ describes its orientation. The course angle $\chi$ is the angle between the direction toward the zenith and the velocity vector of the center of mass of the kite as projected on the tangential plane touching the position of the kite on the half sphere (Fechner, 2016; Fechner and Schmehl, 2018).

bilize the kite using the PID controller. This mathematical model was derived based on the turn rate law (Erhard and Strauch, 2013).

## 2.1  Kinematic framework

₅ As mentioned in the introduction, there are different concepts to derive the mathematical model of the kite (Diehl, 2001; Ahmed, 2014; Fagiano, 2009; Furey, 2012; Thorpe, 2011; Zgraggen, 2014). Some of these models considered the kite as a point mass model, and other researchers just considered ₁₀ the kite as a rigid body (Thorpe, 2011; Zgraggen, 2014; Fechner et al., 2015; Williams et al., 2007). These models are not totally accurate compared to the 10 point mass model (Furey, 2012). However, increasing the number of mass points makes the solver slower than the real flight test. Thus, choosing ₁₅ a simple model to derive the governing equations of the kite potentially allows the solver to run in real time. Additionally, it would be suitable for designing and simulating the FPC and FPP in real time.

To give a complete definition of the kite model, it is im- ₂₀ portant to introduce the different frames used in the derivation of the mathematical model of the kite. The first frame is called the "Earth Centered–Earth Fixed", and the position of the kite and the ground station are measured there. These measurements have to be converted into the "wind reference ₂₅ frame" as shown in Fig. 2. Additionally, the $x_w$ axis of the wind reference frame is in the same direction as the average wind speed, and its center is placed at the anchor point of the

tether. Together with the tether length $l_t$, the elevation angle $\beta$ and the azimuth angle $\phi$ represent a spherical coordinate system that fully defines the position of the kite in the wind ₃₀ reference frame. Figure 2 also introduces the "Small Earth" analogy in which the kite's position is projected onto the unit sphere around the origin and then described by angles $\beta$ and $\phi$.

Based on the given frames of the kite system, the given ₃₅ ~~vectors~~ axes $x_k$, $y_k$, and $z_k$ show the body fixed reference frame of the kite. The $z_k$ axis goes downward from the position of the kite to the connecting point with the tether. The $y_k$ axis is the vector from the left to the right tip of the kite. The $x_k$ axis is the orthogonal ~~of~~ $y_k$ and $z_k$. The heading angle $\psi$ is the angle ₄₀ between the ~~vector~~ axis $x_k$ and the direction toward the zenith as projected on the tangential plane touching the position of the kite on the half sphere. In the model given in this section, the tether is assumed to be straight, and the design of the FPP and the FPC assumed this as well. To control the motion ₄₅ of the kite, the heading angle and the course angle of the kite must be controlled from one point to another using the control action given by the steering motor.

To control the system using the classical control, the system should be converted into a single input single out- ₅₀ put (SISO) model (Baayen and Ockels, 2012), which was achieved after introducing the small Earth reference frame. The input for the kite became the steering action generated from the motor in the kite control unit (KCU), and the system output is the course angle. The angular velocity $\omega$ of the ₅₅

kite point with respect to the origin TS5$O$ is derived in Eq. (1) from the rate of change of the elevation and azimuth angles. This point is further highlighted in Appendix A.

$$\omega = \sqrt{\dot{\beta}^2 + \dot{\phi}^2 \sin^2 \beta} \qquad (1)$$

5  The simplified 2-D kite system model is used in the mathematical equations of the kite. The kite is considered to have a fixed length tether, which has to be straight. The test also considered the kite to be flying toward the center of the wind window in the direction of the zenith. The inputs of the model 10 are the steering action $u_s$, the apparent wind speed $v_a$, and the initial values of the heading, elevation, azimuth, and angular speed. The outputs of the system are the heading angle, its derivative, and the position of the kite, which can be calculated from the elevation and the azimuth angles. The turn 15 rate law is used to calculate the heading angle of the kite, as shown in Eq. (2). After obtaining the heading angle, the integration can easily obtain the value of the heading of the kite. This part is further highlighted in Appendix B.

$$\dot{\psi} = c_1 v_a (u_s - c_o) + \frac{c_2}{v_a} \sin \psi \cos \beta \qquad (2)$$

20  The position of the kite can be calculated from the substitution of the angular velocity $\omega$ into the derivatives of the elevation and the azimuth angles. This concept is valid under the assumption of the similarity of the course angle with the heading angle. Some assumptions are considered when 25 calculating the angular speed $\omega$. One of these assumptions considers that $\omega$ depends only on the elevation angle $\beta$, and it is calculated from Eq. (3); this part is further highlighted in Appendix A.

$$\omega = \frac{\beta_{max} - \beta}{\beta_{max} - \beta_{min}} \omega_{22} \qquad (3)$$

30  ## 2.2 Flight path planner (FPP)

Designing the FPP mainly depends on previous ordered positions that show the required flight path of the kite and the points that the kite should be steered toward. In the work presented in this paper, there are two points called attractor 35 points, on the right and left sides of the wind window, to make the kite fly in a figure eight motion. The figure eight shape was chosen for different reasons: it gives the kite the chance to fly over the wind window to produce more power, by increasing the relative wind velocity. It also aids in smooth 40 steering and reduces the overlapping that occurs if a circular motion is used.

Figure 3 shows the main points of the kite movement. The flight path controller FPC guides the kite to go toward the points set in Fig. 4 in the shape of a figure eight. The algo-45 rithm of the FPP is divided into four subsystems as shown in Fig. 4. The cases of the flight are shown in Table 1 to switch

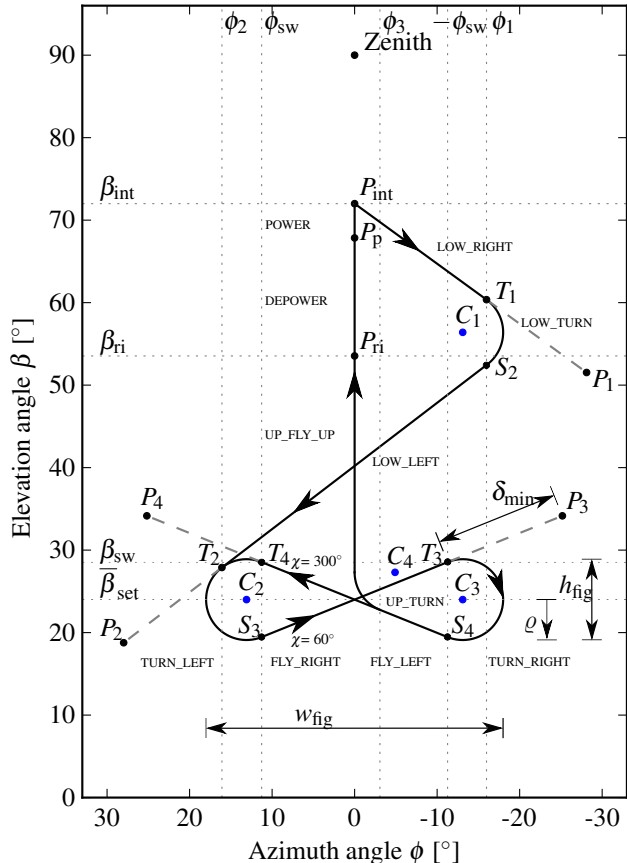

**Figure 3.** Four-step flight path planner for flying a figure eight: First turn left, then steer toward $P_3$, then turn right and finally steer toward $P_4$ (Fechner, 2016; Fechner and Schmehl, 2018).

between the different conditions of flight and sub-states. During turning, there is a time delay, and an offset $\delta_x = 112°$ must be used to compensate for it[1].

To design the FPP, we need to define the inputs and the out-50 put for the algorithm. The kite orientation $\psi$, azimuth angle $\phi$ and set value of the average elevation $\beta_{sw}$ are considered as inputs for the FPP. The control action obtained from the PID controller, set value of the position $P_{k, set}^{SE}$ and set value of the turn rate law $\dot{\psi}_{set}$ are considered as the output. 55

The FPP algorithm needs to obtain the values of $P_3$, $P_4$, and $\dot{\psi}_{turn}$ as a function of the angular width $\omega_{fig}$, the angular height $h_{fig}$, and the minimum attractor point distance $\delta_{min}$.

As shown in Fig. 5, the tangential velocity of the kite $V_{k,\tau}$ is given in Eq. (4): 60

$$V_{k,\tau} = r\omega. \qquad (4)$$

---

[1]This offset is needed to compensate for the time delay between the command to stop turning and the kite actually stopping. This value depends mainly on the rotational inertia of the kite but also on the speed of the steering actuators

**Table 1.** Finite sub-states of the figure-eight flight path planner.

| State | Next state | $P_{k,set}^{SE}$ | $\dot{\chi}_{set}$ | Condition |
|-------|-----------|------------------|--------------------|-----------|
| Initial | TURN_LEFT | – | $\dot{\chi}_{turn}$ | Always |
| FLY_LEFT | TURN_LEFT | – | $\dot{\chi}_{turn}$ | $\phi > \phi_{sw}$ |
| TURN_LEFT | FLY_RIGHT | $P_3$ | from PID | $\chi > 270° - \delta_\chi$ |
| FLY_RIGHT | TURN_RIGHT | – | $-\dot{\chi}_{turn}$ | $\phi < -\phi_{sw}$ |
| TURN_RIGHT | FLY_LEFT | $P_4$ | from PID | $\chi < 90° + \delta_\chi$ |

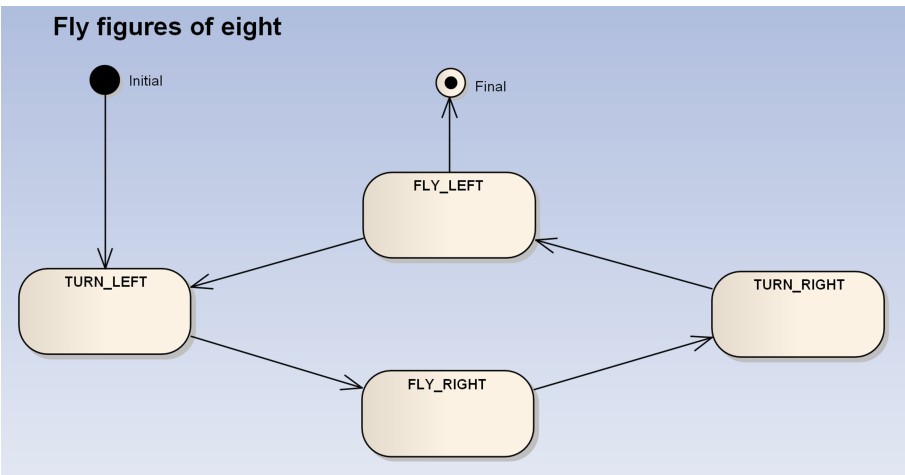

**Figure 4.** Finite sub-state diagram showing the sub-state and the transitional condition of the figure eight controller. This sub-state machine is active in the state FIG-8 of the high-level controller. The states LAST-LEFT and LAST-RIGHT are omitted for simplicity (Fechner, 2016).

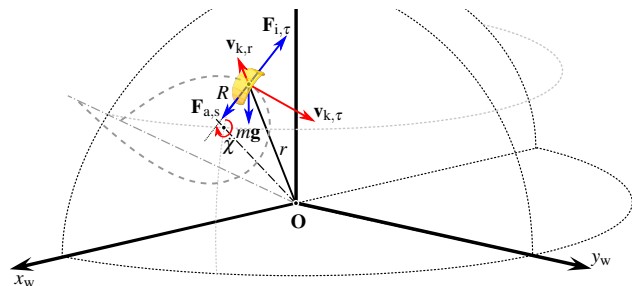

**Figure 5.** Schematic to show the turn rate law of the kite as a function of the angular velocity and turn radius (Fechner, 2016; Fechner and Schmehl, 2018).

Then, the radius of the turn $\varrho$ is given in Eq. (5):

$$\varrho = \frac{h_{fig}}{2}. \tag{5}$$

The turn rate $\dot{\chi}_{turn}$ is calculated from the angular velocity of the kite $\omega$, which is calculated from Eq. (1), and can be calculated as shown in Eq. (6):

$$\dot{\chi}_{turn} = \frac{V_{k,\tau}}{R} = \frac{\omega r}{R}. \tag{6}$$

The value of $\phi_{c2}$ can be calculated from Eq. (7):

$$\phi_{c2} = \frac{w_{fig}}{2} - \varrho. \tag{7}$$

Then, the azimuth angles $\phi_{sw}$ and the elevation angle $\beta_{sw}$ of the switch point can be calculated from Eqs. (8) and (9) by combining the right turning circle with the tangent:

$$\phi_{sw} = \phi_{c2} - \frac{\varrho^2}{\phi_{c2}}, \tag{8}$$

$$\beta_{sw} = \sqrt{\varrho^2 - (\phi_{sw} - \phi_{c2})^2} + \beta_{set}. \tag{9}$$

The slope of the line toward $P_4$ can be calculated from Eq. (10):

$$k = \sqrt{\frac{\phi_{c2} - \phi_{sw}}{\phi_{sw}}}. \tag{10}$$

After solving for the points $P_3$ and $P_4$, we can obtain the following:

$$P_3 = \left( -\phi_{sw} - \delta_{min}\sqrt{\frac{1}{1+k^2}} \right); \left( \beta_{sw} + \delta_{min}k\sqrt{\frac{1}{1+k^2}} \right), \tag{11}$$

$$P_4 = \left( \phi_{sw} + \delta_{min}\sqrt{\frac{1}{1+k^2}} \right); \left( \beta_{sw} + \delta_{min}k\sqrt{\frac{1}{1+k^2}} \right). \tag{12}$$

## 2.3 Flight path controller (FPC)

The kite's position can be controlled using the setting values of the elevation, the azimuth and the normalized depower setting $u'_d$ as inputs and the steering of the motor $u_s$ as the output. The controller includes the navigator of the kite to estimate the desired heading angle based on the current elevation and required elevation $\beta$ and on the azimuth angles $\phi$. The desired flight direction can be calculated by substituting Eqs. (13) and (14) into Eq. (15). The resultant from Eq. (15) can be compared with the heading angle that comes from the sensor to obtain the error signal used in the control design as an input. Then, the steering action can calculated from the control block as an output $u_s$.

$$y = \sin(\phi_{\text{set}} - \phi)\cos\beta_{\text{set}} \tag{13}$$

$$x = \cos\beta\sin\beta_{\text{set}} - \sin\beta\cos\beta_{\text{set}}\cos(\phi_{\text{set}} - \phi) \tag{14}$$

$$\chi_{\text{set}} = a\tan 2(-y, x) \tag{15}$$

After obtaining the result of Eq. (15), the system is ready for the controller, as the SISO and PID controllers were used to update the steering value of the motor $u_s$. LSE and fuzzy control had been described in Sects. 3 and 4 and then the results of fuzzy control were compared with the classical control in Sect. 5 using Simulink[2].

## 3 System identification using least square estimation

The aim of this section is to identify the variation of the system parameters during flight. The parameters must be updated in real time by analyzing the history of the model's input (control action) and output data (course angle). Least square estimation (LSE) (Plackett, 1950; Bobál et al., 2006; Dutton et al., 1997) is used as a system identification technique to update the system's governing equations. The algorithm minimizes the mean square error (MSE) as defined in Eq. (16):

$$\text{MSE} = \frac{1}{k}\sum_{r=1}^{k}(Y_r - Y_m)^2, \tag{16}$$

where $r$ is number of the time steps in the discrete time process, $Y_m$ is the measured data obtained from the sensor, and $Y_r$ is the value results estimated from the system identification shown in Eq. (22). The open loop transfer function of the kite is derived in Fechner (2016) in the form of a simple

---

[2]Simulink is commercial software developed by MathWorks. It is a graphical programming tool for different aspects of engineering. However, it is used here to represent the system's model and design the controller for a fixed sample time. It mainly aims to save time for the user by replacing long code with simple blocks to achieve the same requirements. Simulink is widely used in automatic control and digital signal processing (Reedy and Lunzman, 2010)

model. It has the unknown apparent wind speed as a parameter. Furthermore, the model parameters depend on the angle of attack of the kite, which varies. Thus, the mathematical model cannot exactly define the system (Jehle and Schmehl, 2014b). Therefore, it is suggested to use parameter estimation to update the values for the open loop transfer function, as shown in Fig. 6.

This figure shows that the system is SISO with the course angle required as an input, and the output is the measured course angle of the kite. Then, the error would be calculated from the difference between the input and measured course angle obtained from the sensors. Then, the error signal will be the input for the controller block (adaptive controller) to obtain the suitable control action.

The system identification block will use the control action results from the controller block and the measured course angle as input and then begin estimating the system's parameters (Eq. 23) in real time; these parameters will be used to generate the open-loop transfer function of the kite. These parameters will then be sent to the controller block for use in designing the adaptive control.

This algorithm has the advantage of quickly obtaining system parameter values and has no singularity for any initial conditions, even if they are zeros. The LSE uses the motor action $u_s$ and the sensor data for the course angle to update its parameters. The open loop discrete transfer function for the kite can be approximated as shown in Eq. (17):

$$G(z^{-1}) = \frac{Y(z^{-1})}{U(z^{-1})} = \frac{B(z^{-1})}{A(z^{-1})}. \tag{17}$$

Both the first and second order polynomials would be sufficient to identify the system parameters because the sample time is short, which helps to overcome the error from discretization.

$A(z^{-1})$ and $B(z^{-1})$ are considered as second order polynomial equations in the discrete domain. Thus, the parameters $a_1$, $a_2$, $b_1$, and $b_2$ are the non-dimensional independent variables of the polynomial equations, and they varying with time due to the change that occurs in the system governing equations. After rewriting $A(z^{-1})$ and $B(z^{-1})$ in the discrete form, they are as given in Eqs. (18) and (19):

$$A(z^{-1}) = 1 + a_1 z^{-1} + a_2 z^{-2}, \tag{18}$$

$$B(z^{-1}) = b_1 z^{-1} + b_2 z^{-2}. \tag{19}$$

After substituting Eqs. (18) and (19) into Eq. (17):

$$\frac{Y}{U} = \frac{b_1 z^{-1} + b_2 z^{-2}}{1 + a_1 z^{-1} + a_2 z^{-2}}. \tag{20}$$

Additionally, we can rewrite Eq. (20) in the difference form as shown in Eq. (21):

$$Y_k = -a_1 Y_{k-1} - a_2 Y_{k-2} + b_1 U_{k-1} + b_2 U_{k-2}. \tag{21}$$

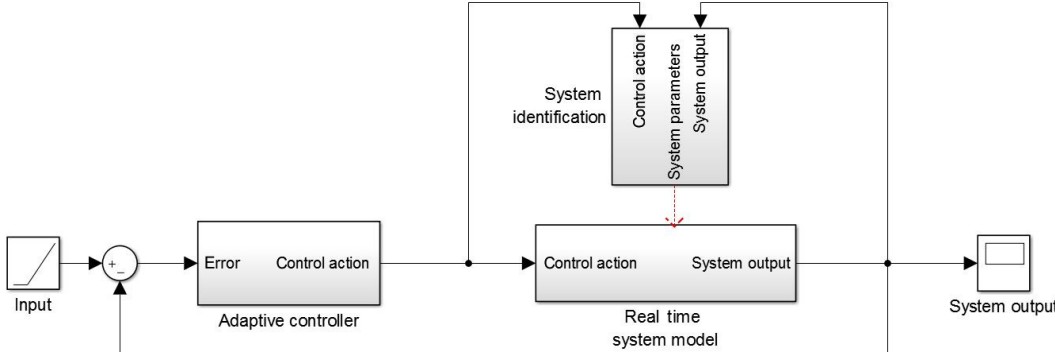

**Figure 6.** Block diagram of the adaptive control system.

Thus, the open loop model can be rewritten as shown in Eq. (22):

$$Y_k = \mathbf{X}_{k-1}^T \boldsymbol{\theta}_k, \tag{22}$$

where:

$$\boldsymbol{\theta}_{k-1} = \begin{bmatrix} -a_1 \\ -a_2 \\ b_1 \\ b_2 \end{bmatrix}, \tag{23}$$

$$\mathbf{X}_{k-1} = \begin{bmatrix} Y_{k-1} \\ Y_{k-2} \\ U_{k-1} \\ U_{k-2} \end{bmatrix}. \tag{24}$$

Thus, the MSE can be written as follows:

$$\text{MSE} = \frac{1}{k} \sum_{r=1}^{k} \left( \mathbf{X}_{r-1}^T \boldsymbol{\theta}_{r-1} - (Y_{\mathrm{m}})_r \right)^2. \tag{25}$$

The objective of using the system identification is to obtain the values of $\boldsymbol{\theta}$ that can minimize the mean square error. From the derivation, the values of $\boldsymbol{\theta}$ can be easily calculated using Eq. (26):

$$\boldsymbol{\theta}_k = \mathbf{P}_k \left[ \sum_{r=1}^{k} (\mathbf{X}_{r-1} Y_{\mathrm{m}}) \right], \tag{26}$$

where,

$$\mathbf{P}_k = \left[ \sum_{r=1}^{k} \left( \mathbf{X}_{r-1} \mathbf{X}_{r-1}^T \right) \right]^{-1}. \tag{27}$$

By rewriting Eq. (12), we can find the following Eq. (28):

$$\mathbf{P}_k = \mathbf{P}_{k-1} - \frac{\mathbf{P}_{k-1} \mathbf{X}_{k-1} \mathbf{X}_{k-1}^T \mathbf{P}_{k-1}}{1 + \mathbf{X}_{k-1}^T \mathbf{P}_{k-1} \mathbf{X}_{k-1}}. \tag{28}$$

After substituting Eq. (28) into Eq. (26):

$$\boldsymbol{\theta}_k = \boldsymbol{\theta}_{k-1} - \frac{\mathbf{P}_{k-1} \mathbf{X}_{k-1}}{1 + \mathbf{X}_{k-1}^T \mathbf{P}_{k-1} \mathbf{X}_{k-1}} \left( \mathbf{X}_{k-1}^T \boldsymbol{\theta}_{k-1} - Y_{\mathrm{m}} \right). \tag{29}$$

Thus, the unknown parameters $a_1$, $a_2$, $b_1$ and $b_2$ should be calculated in every time step. To obtain these parameters, it the following calculation steps must be conducted:

1. Initialize matrix $\mathbf{P}_{k-1}$ with large positive numbers on the leading diagonal and zeros on the off-diagonal elements. The vector $\boldsymbol{\theta}_{k-1}$ must be populated with initial parameters close to the model.

2. $\mathbf{X}_k$ is updated every for sample time by the system outputs and inputs as previously defined.

3. Calculate $\boldsymbol{\theta}_k$ and $\mathbf{P}_k$ from Eqs. (28) and (29).

4. Update $\boldsymbol{\theta}_{k-1}$ and $\mathbf{P}_{k-1}$ with $\boldsymbol{\theta}_k$ and $\mathbf{P}_k$.

5. Repeat the loop for each time step.

Using LSE is a good choice to identify the kite parameters compared with other system identification algorithms. Since it is a non-iterative technique with low computational costs, it has no singularity in the solution, even if the initial conditions are zeros, due to its simple implementation. The results from the LSE are used to predict the behavior of the system (open loop transfer function), and it was used to design the fuzzy control (Sect. 4).

## 4   Fuzzy control

In this section, the control strategy is detailed using Mamdani's fuzzy algorithm (Burns, 2001; Amindoust et al., 2012). Fuzzy logic control (Zadeh, 1968, 1978; Deif et al., 2014) is a digital control technique that uses the multivalued logic output to obtain the solution. It was developed for the systems that do not have accurate mathematical models. Thus, choosing the parameters of the fuzzy controller depends on the experience and the common sense of the designer to overcome the inaccuracy of the mathematical model (Burns, 2001).

The computations of fuzzy control were calculated as hardware-in-the-loop (HIL) (Bondoky et al., 2017). This

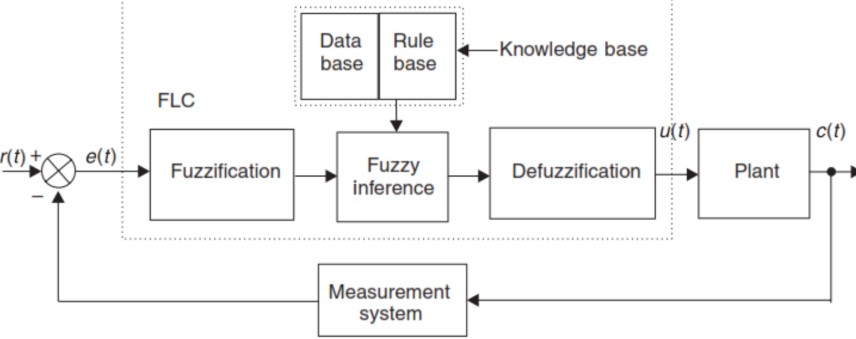

**Figure 7.** Fuzzy logic control system.

means that sensors sent data to the ground station using wireless communications. Then, the calculations were performed using computers on the ground based on the results of the system identification algorithm to choose the suitable control action. Finally, the control action was sent again to the motor to steer the kite.

The kite system consists of an inflatable wing, and its shape changes with time due to the force distribution on its surface. Thus, the mathematical model of the kite cannot be fixed during the whole flight. Moreover, the wind speed varies during the flight, and there is no accurate way to assess it in real time to calculate the force distribution on the kite's surface (van der Vlugt et al., 2013).

Due to all these difficulties, the need for robust control such as fuzzy control to stabilize the kite is very important. Therefore, choosing the fuzzy logic controller is a good choice to satisfy these requirements because it is strong in stabilizing nonlinear systems and can address systems with inaccurate mathematical models. However, the fuzzy logic controller is difficult to implement on small-sized commercial microcontrollers since it requires many calculations that are difficult to implement on microcontrollers fixed on the kite's surface. Therefore, sending the sensor data to the ground station by wireless communications and performing the calculation using a ground station is a good choice to obtain the control action. This step causes a delay due to the transmission time, which is considered in the model and calculation.

Mamdani's model consists of three stages to stabilize the kite system, including fuzzification (Sect. 4.1), inference (Sect. 4.2) and defuzzification (Sect. 4.3), as shown in Fig. 7. The mathematical model used for the simulation was built in TU Delft and given in Fechner (2016). It details the kite model and the flight path controller using classical control $u_s$. Based on the error signal $e$, the input of the fuzzy model can be estimated. Then, the number of memberships will be chosen, and the width of each membership will be changed to tune the system to obtain the suitable control action. The sample time of the simulation plays a very important role in the stability of the kite. Therefore, it should be chosen

based on the hardware used and the speed of calculation in the ground station. In our simulation, the sample time was 0.02 s.

## 4.1 Fuzzification

The process arranges the inputs of the fuzzy logic control to obtain the fuzzy set membership values in the various input universes of discourse (Burns, 2001; Yen and Langari, 1999). To construct the fuzzification stage, one must choose the number of inputs, the size of the universes of discourse and the number and shape of the fuzzy sets. The fuzzy logic control that acts as a proportional controller aims to minimize the error $e$. Therefore, the range of expected values of error $e$ should be known during the estimation of the size of the universes of discourse. In our case, the range of the error $e$ is $-5$ to 5 rad, as shown in Eq. (30)[3]. The last step in designing the fuzzification is to choose the number and shape of fuzzy sets in a particular universe of discourse. Choosing them affects the accuracy of the control action, but it reduces the real time computational complexity. In the simulation, three sets were selected to satisfy the requirements within the given limits, as given in Fig. 8a and b. There was an optimization between the number of sets and the response's accuracy. Therefore, choosing three sets satisfies the stability requirements.

$$e = \begin{cases} \text{trap.}(-8.5, -5, -3, 0) \\ \text{tri}(-1.5, 0, 1.5) \\ \text{trap}(0, 3, 5, 8.5) \end{cases} \tag{30}$$

## 4.2 Rule base and interface

This is the second stage of the fuzzy logic algorithm. It consists of "if-statements" (Burns, 2001) and follows conditional linguistic rules. For example, if $e$ is $N_e$, then $u$ is $N_u$.

---

[3]The range of the error was estimated based on the error of the classical control in Sect. 2.3. Moreover, the tuning for the memberships' shape was estimated from trial and error to obtain a reasonable response for the system.

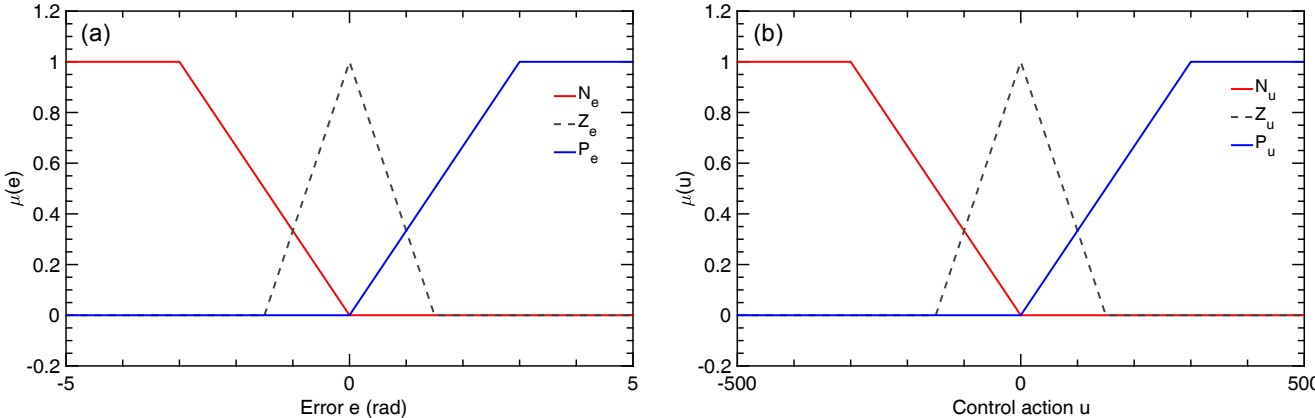

**Figure 8.** The main membership of fuzzification and defuzzification. **(a)** Three set fuzzy input window for error $e$. **(b)** Three set fuzzy output window for control signal $u$.

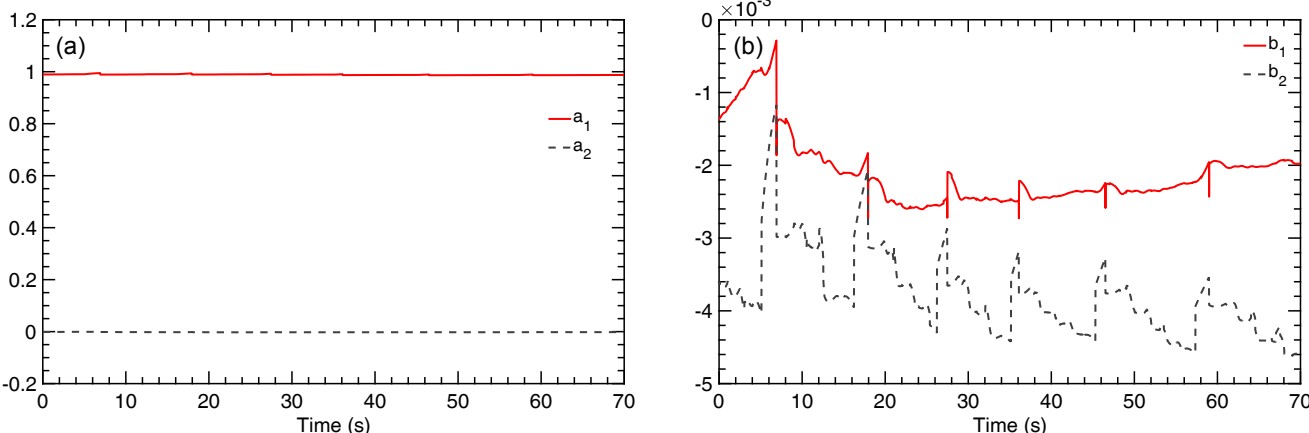

**Figure 9. (a)** Time history of the values $a_1$ and $a_1$. **(b)** Time history of the values $b_1$ and $b_2$.

This style of fuzzy logic control is called the Mamdani rule. Choosing the rule base of the fuzzy logic control depends on the designer's experience with the system. The designer of the rule bases chooses them based on the mathematical model of the system. From the experience of the kite system, the rule bases are chosen as follows:

if $e$ is $N_e$, then $u$ is $N_u$,

if $e$ is $Z_e$, then $u$ is $Z_u$.

Additionally,

if $e$ is $P_e$, then $u$ is $P_u$.

Now the system is ready for the last stage of the fuzzy logic control to obtain the control action.

### 4.3   Defuzzification

This is the last stage of the fuzzy logic control. It is the process of converting the set of inferred fuzzy signals chosen from the fuzzy output, as mentioned in the rule base (Sect. 4.2), into the non-fuzzy (crisp) control action (Deif et al., 2014; Burns, 2001), as shown in Fig. 8a. The most known defuzzification technique is the center of area method. In this case, the control action can be easily obtained by calculating the sum of the first moments of the area divided by the sum of the area. The Matlab fuzzy toolbox is used to simplify the work and save programming time.

## 5   Simulation results

This section shows the result of the system identification (Sect. 3) and the fuzzy control (Sect. 4). The system identification model gives us the definition and description of the kite. The parameters are updated in real time and help us gain the experience needed to design the controller. Fuzzy control was simulated, and the three sets were chosen for the error $e$ and control action $u_s$. The following simulated results were achieved using the model developed in TU Delft (Fechner, 2016). This model gives a detailed description of the kite us-

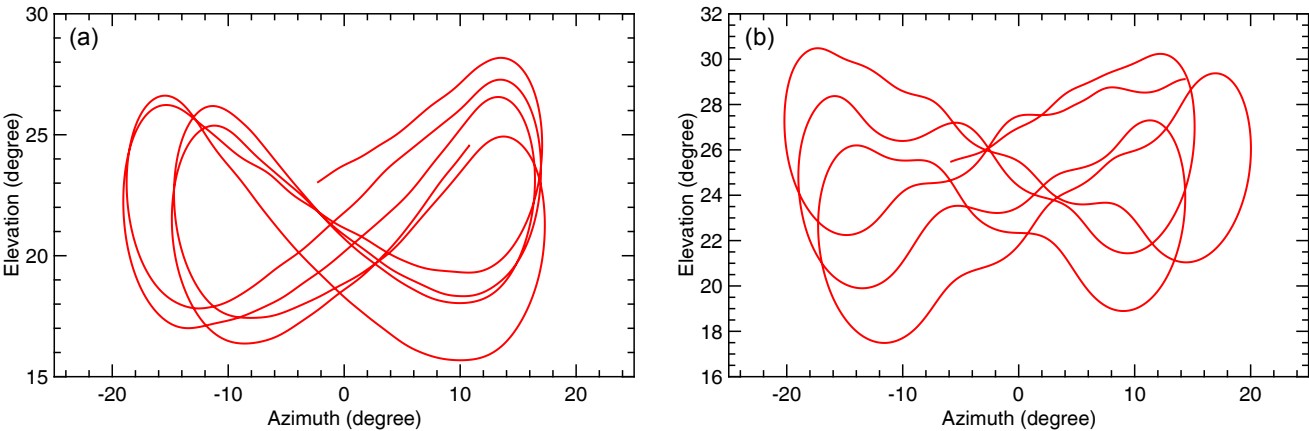

**Figure 10. (a)** Simulation results for the azimuth and elevation angles using classical control. **(b)** Simulation results for the azimuth and elevation angles using SI with fuzzy control.

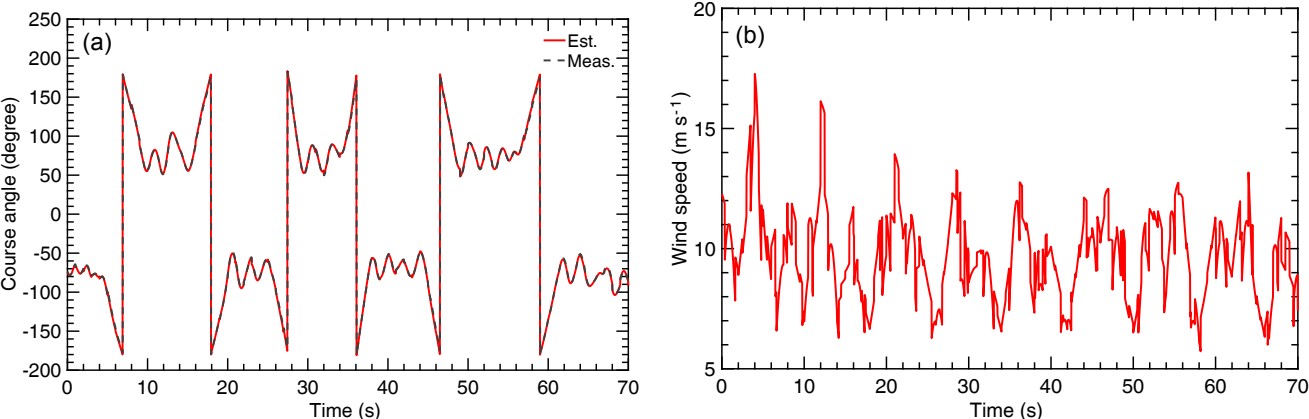

**Figure 11. (a)** Time history of the measured and estimated course angles. **(b)** Time history for the wind speed during the first flight condition.

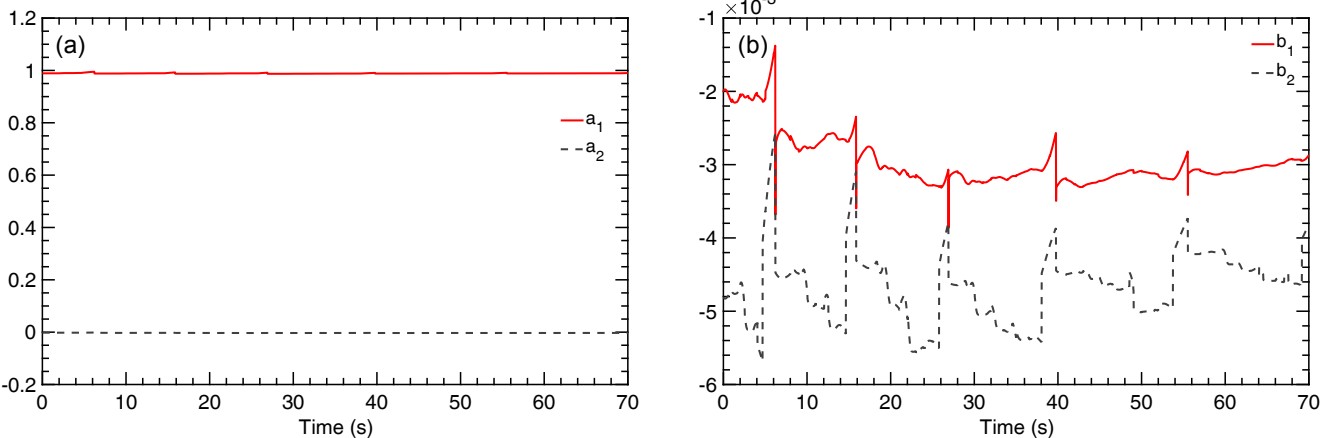

**Figure 12. (a)** Time history of the values $a_1$ and $a_2$. **(b)** Time history of the values $b_1$ and $b_2$.

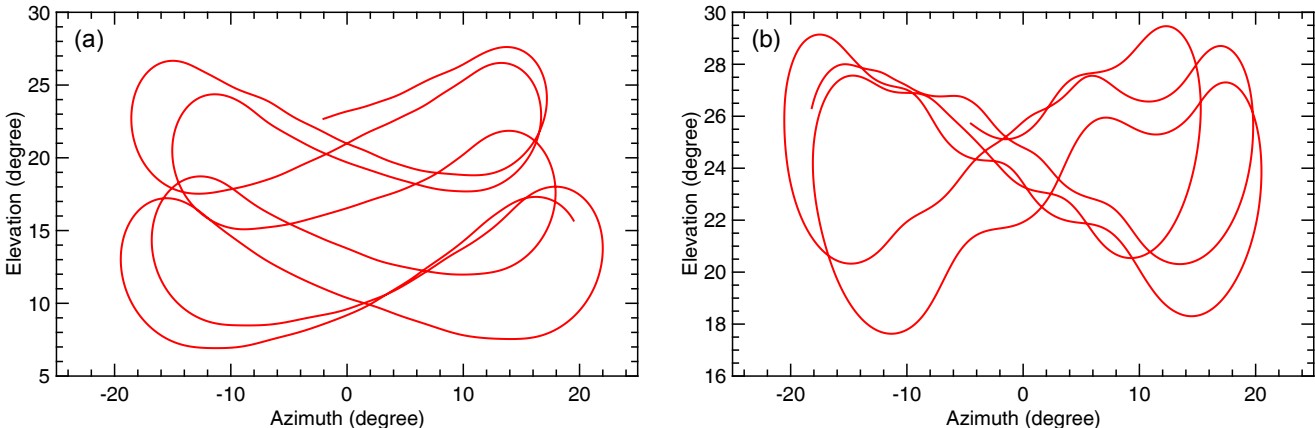

**Figure 13. (a)** Simulation results for the azimuth and elevation angles using classical control. **(b)** Simulation results for the azimuth and elevation angles using SI with fuzzy control.

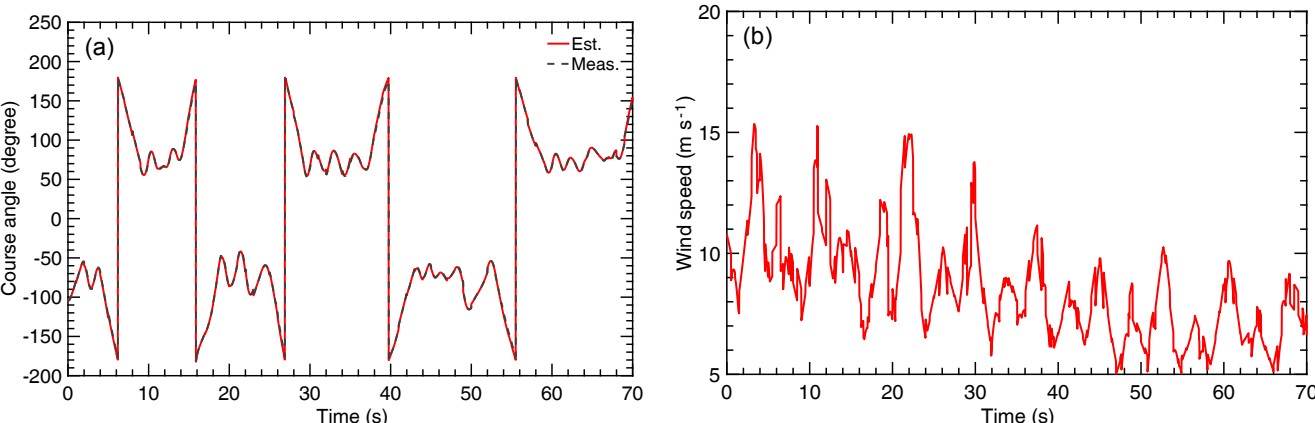

**Figure 14. (a)** Time history of the measured and estimated course angles. **(b)** Time history for the wind speed during the second flight condition.

ing the simple model algorithm, and the flight path of the kite makes a figure eight pattern.

Two flight conditions were tested in this simulation. The difference between the two flight conditions is the wind speed. The wind speed is modeled as shown in Figs. 11b and 14b. The difference between the two models is that the frequency of the wind in Fig. 14b is much higher than that in Fig. 11b. Gaussian noise was added to the sensor data (elevation, azimuth, and apparent wind speed).

### 5.1   Flight condition I

In the first flight condition, the kite model was affected by the wind speed given in Fig. 11b. Thus, the kite's parameters $a_1$, $a_2$, $b_1$ and $b_2$ could be calculated from Sect. 3, as given in Fig. 9a and b. After obtaining the kite's parameters $a_1$, $a_2$, $b_1$ and $b_2$, we can easily compare the course angle of the classical model and the estimated model, as shown in Fig. 11a. The comparison between the figure eight motion is given in

Fig. 10a for the classical control and Fig. 10b for the fuzzy control.

As mentioned in Sect. 4, the fuzzy control will stabilize the kite based on the error signal that comes from the sensors and the input. Thus, it takes the suitable control action to satisfy the requirements.

### 5.2   Flight condition II

In the second flight condition, the wind speed was changed as given in Fig. 14b. The wind speed is modeled to be more aggressive for the kite's controller. This is achieved by increasing the frequency of the wind speed in the first flight test (Sect. 5.1) compared to the second flight test (Sect. 5.2). After applying the system identification algorithm given in Sect. 3, the values of $a_1$, $a_2$, $b_1$ and $b_2$ will be updated as shown in Figs. 12a and b. The figure eight motion given in Fig. 13b is calculated using the simple model and the classical controller. The figure eight concept is satisfied, but the

elevation angle is reduced toward the instable region. Thus, using the classical control cannot satisfy the condition of stability in different wind conditions. However, Fig. 13a is calculated using the fuzzy control, which can handle the strong changes in the wind speed in addition to the noise that comes from the sensors using the same algorithms without any change in the code. The comparison between the course angles measured and estimated using the system identification are given in Fig. 14a. Even though the wind speed was changed, the system identification can predict the course angle to become almost identical to that measured from the sensors.

## 6 Conclusions

This paper presented a technique to identify the kite's parameters and controller that would be robust enough to stabilize the kite in real time when other classical controls cannot satisfy this. Using the least square estimation algorithm for system identification helps to present a complete definition for the kite's parameters in real time. The variation of the kite's parameters comes from the changes in wind speed and direction, the change in the aerodynamic coefficients, and the change in the kite's shape (as it consists of an inflatable wing).

The kite model is mainly non-linear. Therefore, the choice of fuzzy control is suitable for such systems. Additionally, the computations of fuzzy control were calculated as HIL. When deriving the system identification equations, the model was considered as a discrete linear model with a short sample time. The results of the system identification were compared with the classical model for different wind speeds, as shown in Figs. 9–14, which show the differences between the classical and fuzzy controls in stabilizing the kite.

**Data availability.** 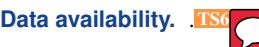

## Appendix A: The angular velocity $\omega$ of the kite motion

The kite is moving in two directions (spherical coordinates). $\phi$ is the rotation around the $Z_w$ axis, and $\beta$ is moving in the direction of $X_{se}$, as shown in Fig. 2. They can be as given in Eq. (A1):

$$\boldsymbol{\omega} = (\dot{\phi})\boldsymbol{z_w} + (\dot{\beta})\boldsymbol{y_{se}}, \tag{A1}$$

where $z_w$ is given in Eq. (A2).

$$\boldsymbol{z_w} = -(\sin\beta)\boldsymbol{z_{se}} \tag{A2}$$

From Eq. (A1) and Eq. (A2), we obtain the following:

$$\boldsymbol{\omega} = -(\dot{\phi}\sin\beta)\boldsymbol{z_w} + (\dot{\beta})\boldsymbol{y_{se}}. \tag{A3}$$

The angular velocity $\omega$ of the kite point with respect to the origin $\boldsymbol{O}$ is derived in Eq. (A4) from the rate of change of the elevation and azimuth angles.

$$\omega = \sqrt{\dot{\beta}^2 + \dot{\phi}^2\sin^2\beta} \tag{A4}$$

Another simplified equation is given as a function of the elevation angle $\beta$ in Eq. (A5). This equation is derived from an experimental test, and the relationship between the angular speed and elevation angle is assumed to be linear within a specified range for the elevation angle $\beta$.

$$\omega = \frac{\beta_{max} - \beta}{\beta_{max} - \beta_{min}}\omega_{22} \tag{A5}$$

This equation assumes that the angular speed reaches the constant value $\omega_{22}$ at an elevation angle of 22° and an angular speed of zero at $\beta = \beta_{max} = 73°$. This angle is the elevation angle of the Hydra kite while parked at a 300 m tether length for an approximate wind speed of $6\,\mathrm{m\,s^{-1}}$. From the experimental test for the Hydra kite used in the simulation, we obtained the values of $\beta_{max} = 73°$, $\omega_{22} = 0.25°/s$ and $\beta_{min} = 22°$ (Fechner, 2016).

## Appendix B: Turn rate law

The law states that the turn rate of the kite about its yaw axis is a function of the steering deflection of the actuator $u_s$, the kite's dependent constants $c_o$, $c_1$, $c_2$, the heading angle $\psi$, the elevation angle $\beta$ and the apparent wind speed $v_a$ (Erhard and Strauch, 2013). The turn rate law is used to calculate the heading angle of the kite as shown in Eq. (B1). After obtaining the rate of the heading angle, the integration can easily obtain the value of the heading of the kite.

The steering value of the motor $u_s$ is the control action responsible for steering the kite. It is the change of the length of the tether connected between the kite and the KCU. The control action $u_s$ is calculated based on the calculations of the FPP and FPC.

$$\dot{\psi} = c_1 v_a(u_s - c_o) + \frac{c_2}{v_a}\sin\psi\cos\beta \tag{B1}$$

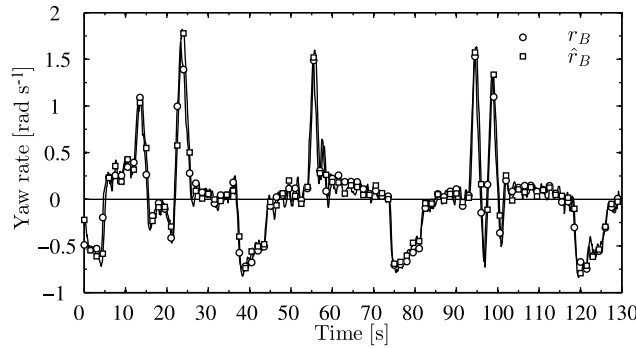

**Figure B1.** Comparison between the estimated turn rate and the measured turn rate for the $25\,\mathrm{m}^2$ kite surface area (Jehle and Schmehl, 2014b).

**Table B1.** ⬛TS7 Fitted turn rate law parameters of the Hydra kite (Fechner, 2016).

| Parameter | Measured | 1 p model | 4 p model |
|---|---|---|---|
| $u_d$ [%] | 26.0 | 26.0 | 26.0 |
| $c_0$ [–] | −0.003 | −0.004 | −0.003 |
| $c_1$ [$\mathrm{rad\,m^{-1}}$] | 0.261 | 0.264 | 0.262 |
| $c_2$ [$\mathrm{rad\,m\,s^{-2}}$] | 6.28 | 6.20 | 6.27 |
| $\rho$ (PCC) | 0.9933 | 0.9999 | 0.9995 |
| $\sigma$ [$\mathrm{rad\,s^{-1}}$] | 0.002 | 0.0002 | 0.0006 |

The algorithm is an iterative technique to obtain the empirical relationship between the kite parameters $c_1$ and $c_2$ and the turn rate of the kite $\dot{\psi}$. The characteristics of the kite (such as its size and weight) are considered in the parameters $c_1$ and $c_2$, which estimate the main behavior of the kite.

An empirical relationship is achieved in (Jehle and Schmehl, 2014b) for other projects, as shown in Fig. B1, for the $25\,\mathrm{m}^2$ kite surface area, but it is not used in this thesis. From this experiment, the authors obtained the parameters of the kite as $c_1 = 0.153$ and $c_2 = 0.25$. The parameters used in the simulation to substitute in the turn rate law in this thesis are given in Table B1. The Hydra kite with a $10.18\,\mathrm{m}^2$ projected surface area was used in the simulation, and the experiment was implemented to obtain its parameters (Fechner, 2016).

## Appendix C: Nomenclature

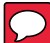

| | | |
|---|---|---|
| $Y(z^{-1})$ TS8 | Estimated course angle obtained from the system identification in $Z$ domain | rad |
| $G(z^{-1})$ | Open loop transfer function of the model in $Z$ domain | – |
| $U(z^{-1})$ | System input which is defined as $u_s$ in $Z$ domain | – |
| $k$ | Summing counter | – |
| $z^{-1}$ | Backward shift operator in $Z$ domain | – |
| $x_w, y_w, z_w$ | Wind reference frame of the kite | – |
| $x_{se}, y_{se}, z_{se}$ | Unit sphere reference frame | – |
| $x_k, y_k, z_k$ | Body fixed reference frame of the kite | – |
| $x'_k, y'_k, z'_k$ | Projection of the body fixed reference frame on the unit sphere | – |
| $h_{fig}$ | Angular height | rad |
| $u_s$ | Steering action | – |
| $c_0$ | Steering offset of the turn rate law | – |
| $c_1$ | Steering sensitivity coefficient of the turn rate law | – |
| $c_2$ | Gravity sensitivity coefficient of the turn rate law | – |
| $v_{w,ref}$ | Horizontal wind velocity at the reference height | $m\,s^{-1}$ |
| $y_r$ | Estimated course angle obtained from system identification | m |
| $y_m$ | Measured course angle obtained from the sensor | m |
| $\chi_{set}$ | Set value for the course angle | rad |
| $\mu$ | Membership function (the range is from 0 to 1) | – |
| $\varrho$ | Turn radius of the trajectory of the kite point | rad |
| $\delta_{min}$ | Minimal, angular attractor point distance | rad |
| $\omega_{fig}$ | Angular width | $rad\,s^{-1}$ |
| $\omega_{22}$ | Angular speed at elevation angle 22° | $rad\,s^{-1}$ |
| $\omega$ | Norm of the angular velocity of the kite on the unit sphere | $rad\,s^{-1}$ |
| $\beta_{min}$ | Elevation angle at angular speed $\omega_{22}$ | rad |
| $\beta_{max}$ | Elevation angle at zero angular speed | rad |
| $\beta$ | Elevation angle | rad |
| $\phi$ | Azimuth angle | rad |
| $\chi$ | Course angle | rad |
| $\psi$ | Heading angle | rad |
| $P_{k,set}^{SE}$ | Position of the kite in angular coordinates ($\phi$, $\beta$) | rad |
| $V_a$ | Apparent wind speed | $m\,s^{-1}$ |
| $P_k$ | Covariance matrix of the estimated error | – |
| $Y_m$ | Measured course angle obtained from the sensor | rad |
| $\hat{\theta}$ | Last vector estimated using the LSE algorithm | – |

**Competing interests.** All authors declare that there is no support from any organization for the submitted work. There are have been no financial relationships with any organizations that might have an interest in the submitted work in the previous three years. There are no other relationships or activities that could have influenced the submitted work.

**Acknowledgements.** Roland Schmehl was supported by the H2020-ITN project AWESCO funded by the European Union's Horizon 2020 research and innovation program under the Marie Skłodowska-Curie grant agreement no. 642682.

Edited by: Joachim Peinke
Reviewed by: two anonymous referees

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

Please note the remarks at the end of the manuscript.

## Remarks from the typesetter

**TS6**    Please provide a statement on how your underlying research data can be accessed. If the data are not publicly accessible, a detailed explanation of why this is the case is required. The best way to provide access to data is by depositing them (as well as related metadata) in reliable public data repositories, assigning digital object identifiers (DOIs), and properly citing data sets as individual contributions. Please indicate if different data sets are deposited in different repositories or if data from a third party were used. If no DOI is available, assets can be linked through persistent URLs to the data set itself (not to the repositories' home page). This is not seen as best practice and the persistence of the URL must be secured.