# Peer review of "System Identification, Fuzzy Control, and Simulation Results for a Fixed Length Tether of a Kite Power System"

_Wind Energy Science, 2017_

## Short Comment (SC1) · 3 Jul 2017

To say AWE follows two different concepts; Massively limits the rich conceptual space openly identified in AWES.
* * *

---

## Referee Comment (RC1) · Anonymous Referee #1 · 15 Sep 2017

This paper considers the system identification and fuzzy control of an airborne energy system based on a tethered kite.

The paper presents a research problem interesting to the community.

However, the paper is not well written. It needs a lot of revisions to be clearer. For example, words like "very" and "thing" are not scientific, "got" should be replaced with "obtained" and so on.

Although the literature survey is comprehensive, it did not weave a good story. It is more of stating who did what than making a case.

[Figure]

The novelty of the paper is not very clear. The system identification algorithm is well known and the paper does not have a strong modeling, experimental, or computational component.

---

## Referee Comment (RC2) · Anonymous Referee #2 · 13 Dec 2017

The topic is definitively unique and the work you have done is remarkable. Nevertheless, the paper does need some work to be considered for a final publication. I would therefore, urge you to consider the following point for an improvement:

1. There are some language issues I have. Sometimes sentences are hard to read, not easy to understand or repeat certain words (like "and") very often. Also always keep articles in mind: It is "The" kite system on line 14 on page 1. Please also check line 8-9 on page 2. Line 10 (ending) on page 2. The "ands" in line 19-20 and the sentences around it on page 2. "The thing that" on line 8 on page 3. "The" average system model on line 13 same page. "Another thing" in line 20 page 3 – which thing? . . . so please

reread the text critically concerning the language and expression!

2. Page 2: Where does the wind energy density range from ... ? Is this globally?? Please cite a source!

3. Page 3 Paragraph lines 14-19, please rephrase a bit and watch the language. It should be formal.

4. Page 3 in the paragraph afterwards: How can a technique be promising, if it is only valid for a non realistic case?

5. Page 3 and further: What is a classical controller? Is this defined somewhere?

6. Define all magnitudes used in the equations in the text! All! Otherwise the rest gets seriously unclear.

7. Page 5 line 11 you mention, that you minimize Jk. How? Where is this mathematically described?

8. In the section 3, please make sure, you cite every source you have been using properly.

9. Equation (17), why did you use these magnitudes? How did you get them?

10. Section 3.3, please refer to some source for further reading. Remember: It is science. This means everything you have been done, needs to be possibly redone by others to check for validity. This should be possible after reading your paper and all the sources you cited.

11. In figures 4, 5, 6, 7, 11, 12: What are the magnitudes on the y-axis? Where is this explained?

12. On page 11 line 2 you mention classical (?) and fuzzy control in the figures. But it doesn't say in the caption of the figures, which is which.

13. Page 11 line 6: More shaper does not exist.

14. Why is in figures 8,9, 14, 15 the path given in degrees? Please give a reason in the text.

15. On page 14 line 5 you mention for the first time "hardware in the loop". What do you mean by this now – as you mention it the first time.

---

## Author Comment (AC1) · 10 Jan 2018

To say AWE follows two different concepts; Massively limits the rich conceptual space openly identified in AWES.

The reviewer is correct in his opinion and the paper was modified to adapt this point (page 3 line: 20) as following:

Power generation by AWE follows several concepts, however, in this paper we mentioned just two different concepts.

---

## Author Comment (AC2) · 10 Jan 2018

This paper considers the system identification and fuzzy control of an airborne energy system based on a tethered kite. The paper presents a research problem interesting to the community. However, the paper is not well written. It needs a lot of revisions to be clearer. For example, words like "very" and "thing" are not scientific, "got" should be replaced with "obtained" and so on.

The paper had been revised and a lot of modifications were added based on the reviewer comments.

Although the literature survey is comprehensive, it did not weave a good story. It is more of stating who did what than making a case.

In the literature survey, I tried to move from the general topic then move to be specific to my problem which I am trying to solve as following:

- General introduction for the AWE (page 3, lines 5-10).
- Presenting the benefits of the AWE from the side of power density (page 3, lines 11-19).
- Describe different concepts for power generation from the AWE systems (page 3, lines 20-26).
- Discuss the previous models and control theories, system identification, robustness of simulations to overcome the uncertainty of the model (page 3, lines 27 - page 5, line 10).
- Work achieved in the presented paper with the novelty of my work (page 5, lines 11-20).
- The structure of the paper (page 5, lines 21-26).

The novelty of the paper is not very clear. The system identification algorithm is well known and the paper does not have a strong modeling, experimental, or computational component.

- This part was rewritten to be more clear for the reader in the end of section (1 Introduction) as following:

(In this paper, the least square estimation (LSE) was used as a system identification to get a more accurate description for the steering dynamics of the kite in real-time; the characteristics of the kite are varying with time because the wing is inflatable and flexible. Also, the wind speed can't be measured in real-time, thus it is impossible to obtain the lift and drag forces during flight. This technique especially is used to identify the system parameters as it can calculate them without iteration (one directional calculations) which means no time loses and low chance of singularity in the solver.

The novelty of this work is to use an algorithm that is valid for any kite size and any tether length. So it can overcome the problems of the uncertainty. The LSE algorithm needs the steering values from the motors and the course angle from the sensors. Thus, no additional information is needed such as the wind speed or the mathematical model of the kite to identify the system that shall be controlled. Therefore, this paper tries to stabilize the kite using fuzzy control based on the LSE in real-time.

This paper is divided into five main sections. The first section is the introduction1, which gives an overall view of the previous research related to the paper's work. The second section shows the mathematical model2used to describe the kite's motion. The third section gives the system identification derivation and details the sequence of the code3. The fourth section describes

the main parts of the fuzzy control and explains the choice of the fuzzy control parameters4. Finally, the last section shows the simulation results of the classical control and the fuzzy algorithm. The comparison also includes varying wind conditions and their effects on system stability5.)

- The modelling for the kite motion was added in section (2 Mathematical model) to make the paper more clearly for the reader.

- In general, this paper just presented the concept of system identification to the kite system supported by simulation and fuzzy control, moreover, a comparison between the classical control (which had been achieved in TU Delft), and the LSE with fuzzy control was presented in this paper to show the reliability of the presented work in different wind conditions.

---

## Author Comment (AC3) · 10 Jan 2018

The topic is definitively unique and the work you have done is remarkable. Nevertheless, the paper does need some work to be considered for a final publication. I would therefore, urge you to consider the following point for an improvement:

1. There are some language issues I have. Sometimes sentences are hard to read, not easy to understand or repeat certain words (like "and") very often. Also always keep articles in mind: It is "The" kite system on line 14 on page 1. Please also check line 8-9 on page 2. Line 10 (ending) on page 2. The "ands" in line 19-20 and the sentences around it on page 2. "The thing that" on line 8 on page3. "The" average system model on line 13 same page. "Another thing" in line 20 page 3 – which thing? . . . so please reread the text critically concerning the language and expression!

The paper had been revised and a lot of modifications were added based on the reviewer comments.

2. Page 2: Where does the wind energy density range from . . . ? Is this globally?? Please cite a source!

This part is rewritten in detail in page. 3 lines (11-20) as following:

AWE systems can capture more energy with higher capacities, which is why they are considered a good renewable energy system. The wind energy density at an altitude of 10 km could reach up to 5000 W/m2 according to Wubbo Ockels, the developer of the "ladder-mill" concept in 1997Ockels(2001). However, it is too hard to build a system that can operate at an altitude of 10 km and generate electricity from wind. This is why most of the current development and research projects have shifted their focus to lower altitudesArcher et al.(2014).
Wind energy density ranges from 1400 to 4500 W/m2 at altitudes of 200 to 900 m, respectively. Wind turbines cannot be installed at these altitudes because of the limitations of tower sizeGoudarzi et al.(2014). Therefore, it will be an optimum solution to have a system similar to wind turbines at this altitude but with no tower. The concept of wind turbine blade rotary motion can be replaced by a tethered kite connected via the flexible wing to a fixed generator on the ground.

3. Page 3 Paragraph lines 14-19, please rephrase a bit and watch the language. It should be formal.

This part is modified to be formal (page 4, lines: 15 – 18) as following:

Experimental efforts for the autonomous take-off of the airborne systems were carried out, however there are still some challenges to get fully autonomous flight in different wind conditions. Moreover, a global controller that can work under all conditions cannot be designed effectively for the commercial products Fechner and Schmehl(2012);Jehle and Schmehl(2014a);Baayen and Ockels (2012).

4. Page 3 in the paragraph afterwards: How can a technique be promising, if it is only valid for a non-realistic case?

The technique is promising in the theoretical simulations to optimize flight trajectory, but in a real flight test, it will require accurate and fast wind data that are currently unavailable (due to restrictions of hardware). Thus, in the future, it is expected to obtain these data so fast and accurate to be used in the real-time flight tests.

5. Page 3 and further: What is a classical controller? Is this defined somewhere?

New section is added (2 Mathematical model); it gives a lot of details to explain the mathematical model and the classic control.

6. Define all magnitudes used in the equations in the text! All! Otherwise the rest gets seriously unclear.

All magnitudes used in the equations are added in the beginning of the paper under the section of (Nomenclature)

7. Page 5 line 11 you mention, that you minimize Jk. How? Where this is mathematically described?

The section of (3 System Identification Using Least Square Estimation) is written in detail and it was modified to be clearer to the reader.

8. In the section 3, please make sure, you cite every source you have been using properly.

More references are added in the section.

9. Equation (17), why did you use these magnitudes? How did you get them?

This part was added in section 4, as given in page 16 lines (8) to page 17 lines (5):

In our case, the range of the error e is -5 to 5 rad, as shown in Eq. (30)[3]. The last step …..
Choosing 3 sets satisfies the stability requirements.

10. Section 3.3, please refer to some source for further reading. Remember: It is science. This means everything you have been done, needs to be possibly redone by others to check for validity. This should be possible after reading your paper and all the sources you cited.

Two sources are added in this subsection (4.3 Defuzzification):

This is the last stage of the fuzzy logic control. It is the process of converting the set of inferred fuzzy signals chosen from the fuzzy output, as mentioned in the rule base4.2, into the non-fuzzy (crisp) control actionDeif et al.;Burns(2001), as shown in Fig.8b. The most known defuzzification technique is the center of area method. In this case, the control action can be easily obtained by calculating the sum of the first moments of the area divided by the sum of the area. The Matlab fuzzy toolbox is used to simplify the work and save programming time.

11. In figures 4, 5, 6, 7, 11, 12: What are the magnitudes on the y-axis? Where is this explained?

All the magnitudes are added in the nomenclatures section; the new numbering for them becomes Fig. 8,9, and 12. The y-axes for figures 9, and 12 are non-dimensional, and figure 8 has a symbol of $\mu$, it was added on the figure; it was a mistake to not write it in the first draft.

12. On page 11 line 2 you mention classical (?) and fuzzy control in the figures. But it doesn't say in the caption of the figures, which is which.

The caption for the classical control and fuzzy control are modified in figures 10 and 13. The difference between the fuzzy and classic controllers are shown as a subfigures for the same figure.

13. Page 11 line 6: More shaper does not exist.

The shape of the wind speed was replaced from the triangle shape to be smoother with changing in the frequencies; it is explained in details (page 18 line 3-5) as following:

(The difference between the two flight conditions is the wind speed. The wind speed is modelled as shown in Figs.11band14b. The difference between the two models is that the frequency of the wind in14bis much higher than that in11b. Gaussian noise was added to the sensor data (elevation, azimuth, and apparent wind speed). )

14. Why is in figures 8,9, 14, 15 the path given in degrees? Please give a reason in the text.

The physical meaning of those figures are mentioned in section (2 Mathematical model). Therefore the reason of using degree was explained in detail.

15. On page 14 line 5 you mention for the first time "hardware in the loop". What do you mean by this now – as you mention it the first time.

The meaning of the Hardware of the loop is explained in page 15 lines (13-16) as following:

The computations of fuzzy control were calculated as hardware-in-the-loop (HIL) Bondoky et al.(2017). This means that sensors sent data to the ground station using wireless communications. Then, the calculations were performed using computers on the ground based on the results of the system identification algorithm to choose the suitable control action. Finally, the control action was sent again to the motor to steer the kite.

---

## Author Response (AR2)

Associate editor's comments:

1. On page 6 line 14: It would be nicer to write … has been achieved in TU Delft "by" Fechner et al..

The paper wad modified as mentioned in the comment.

2. In page 12 line 3, I believe you mean, that you describe the fuzzy control in sections 3 and 4 and compare it in the simulation given in section 5. The way the sentence is written right now, it is not clear.

The paragraph is modified as following: "LSE and fuzzy control had been described in3,4, then the results of fuzzy control were compared with the classical control in5 using Simulink."

3. I'm still a bit picky with the nomenclature: Please define G, Y, U, k and z

All these parameters are modified in the nomenculature section.

[revised manuscript text omitted]